# Preparation of Highly Stable Polymer Microstructure with Enhanced Adhesion Strength by Pushpin-like Nano/Microstructure Array

**DOI:** 10.3390/polym15041015

**Published:** 2023-02-17

**Authors:** Yongjin Wu, Guifu Ding, Yuan Zhu, Yan Wang, Rui Liu, Yunna Sun

**Affiliations:** 1National Key Laboratory of Science and Technology on Micro/Nano Fabrication, Shanghai Jiao Tong University, Shanghai 200240, China; 2School of Electronic Information and Electrical Engineering, Shanghai Jiao Tong University, Shanghai 200240, China; 3National Demonstration Center for Experimental Materials Science and Engineering Education, Jiangsu University of Science and Technology, Zhenjiang 212003, China

**Keywords:** transition strengthening layer, adhesion strength, stable adhesion ability, SU-8 microstructure, universal adhesion-strengthen method

## Abstract

This polymer microstructure expands more available application, which is a milestone for the development of micro-electro-mechanical system devices towards intelligence and multifunction. Poor interface bonding between the polymer and Si or metal is a particular problem, which restricts the application and promotion of polymer materials. In this study, a transition strengthening layer is proposed to obtain a highly stable polymer microstructure by enhancing the interfacial adhesion strength. The transition strengthening layer is activated by a pushpin-like nano/microstructure array with micromachining technology. Given its good graphical qualities and compatibility, epoxy negative photoresist SU-8 is applied to evaluate the strengthened capabilities of the pushpin-like nano/microstructure array. The microstructure of SU-8 is prepared by the same processes, and then the adhesion strength between the SU-8 microstructure and various activated substrates is tested by the thrust tester. It was determined that SU-8 with an activated pushpin-like microstructure array possessed a highly stable adhesion ability, and its adhesion strength increased from 6.51 MPa to 15.42 MPa. With its ultrahigh stable adhesion ability, it has been applied in fabricating three typical microstructures (hollow square microstructure, gradually increasing adjacent periodic microstructure, and slender strip microstructures) and large-area SU-8 microstructures to evaluate the feasibility of the transition strengthening layer and repeatability and universality of the microfabrication processes. The drifting and gluing phenomenon are avoided by this method compared with the traditional design. The proposed pushpin-like nano/microstructure array is promising in enhancing the stability of polymer microstructures with a substrate.

## 1. Introduction

The technology of micro-electro-mechanical systems (MEMS) has extended from a simplified surface structure as the core design in the initial stage to a complex design with a three-dimensional spatial structure and multiple structural materials; polymer microstructures with functional properties are one of the key microstructures to realize function expansion [1,2,3,4,5]. The development of diversified polymer structures means that the structure can obtain more available applications. This expansion is a milestone for the in-depth development of intelligent and multifunctional MEMS devices from a macro perspective. Three-dimensional structures are outstanding in adhesion, photocatalysis, and dynamic characteristics, which will provide a new direction for the further miniaturization of MEMS functional devices. For example, MEMS micro capacitors use height space to store energy to obtain higher energy and power density [6]. Microchannel heat sinks use limited height space for efficient fluid solid heat exchange to further strengthen heat dissipation capacity [7,8]. Polymer structures have potential in flexible devices, optical devices, and biochemical devices due to their high stability, compatibility, transparency, and flexibility. Huang et al. proposed a self-assembled three-dimensional MEMS device based on polyimide (PI), which consists of two surface structures connected by PI joints. A lifting angle of the free surface could be modified up to 74 deg by the varied ratio of PI thickness to plane size and curing temperature [9]. To balance power consumption and visibility of optical biopsy in minimally invasive medical devices, Ribeiro et al. fabricated a tilted polydimethylsiloxane microlens through the hanging droplet approach, achieving 4-fold image amplification and 30% improvement of light-emitting diode illumination irradiance without increasing power consuming [10]. In Lien’s work [11], a flexible thermoplastic polyurethane sheet was used as a substrate for a transparent 360° omnidirectional photodetector, enabling the photodetector device to be stretched and attached to both flexible and curved surfaces, broadening the application fields of optical devices.

Other than their unique properties, polymers also have serious issues during fabrication, including swelling and internal stress [12,13,14,15]. They have great influence on the reliability and adhesion strength of polymers and substrates [16,17,18]. The diversified choice of heterogeneity enriches the application field of MEMS, which is an inevitable trend of development. However, the interface bonding between a polymer and silicon (Si) or metal is poor, especially between a polymer and metal. When using polymer to fabricate microstructures or microdevices on a metal substrate, the adhesion strength between the polymer and substrate is much lower than that on the Si substrate [19], which will have a great impact on the final quality and performance of the designed structure. The polymer and substrate are prone to bond failure due to the poor adhesion performance between the polymer and metal materials. Furtherly, because the thermal expansion coefficient of polymer is different from that of the metal, internal stress is generated in the film. With the increase in the thickness and area of polymer, the internal stress in the film increases, resulting in the debonding of the interface. In serious cases, it will cause the complete damage of the pattern.

In order to solve this issue, a set of studies have been carried out to improve the adhesion between polymers and substrates. Many examples of research focus on the optimization of these process to reduce the internal stress of the film, and then enhance the interface adhesion [17,20,21,22,23,24]. In [25,26,27,28], an adhesive promoter was introduced to increase the adhesion between polymer and substrate. In [29,30], the surface modification of polyester films was achieved via plasma to improve interface adhesion. Morphology modification has proved to be a universal way to improve bonding strength by increasing the contacting area and introducing a mechanical interlock between polymers and substrates, without functional groups affected by corrosion, laser ablation, plasma treatment, and etching [31,32,33]. The formed rough interface mainly increases the contact area and forms one direction interlock. However, there is still room for improvement in the uniform of the roughened morphology and in the interlocking latitude from one or two dimensions to three dimensions. Although the adhesion between the polymer and substrate can be improved by process optimization, additional limited structure, morphology modification, and other methods, its versatility is weak and it is not suitable for all polymer microstructures in widen application.

With high photosensitivity, high thermomechanical stability, good optical transparency, and excellent graphical capability, SU-8 negative photoresist (SU-8) has been used widely for MEMS devices [34,35]; thus, SU-8 is selected as the polymer in this work. Microfabrication technology such as sputtering, electroplating, oxygen ion treatment, etc., are used to texture the nano/microstructure to form a transition strengthening layer. Thus, the sputtered cuprum (Cu) film, electroplating Cu film, and oxygen ion bombardment are considered to enhance the interfacial adhesion by affecting the grain size and surface topography of the substrate. Further, a transition strengthening layer is constructed to improve the interfacial adhesion since it can not only enlarge the contacting interfacial surface area of the SU-8 and the treated substrate, but also introduce the fasten area. To ensure the accuracy of the adhesion strength comparison between different interfaces, the SU8 micro-pillars used for testing are fabricated on the treated surfaces with the same procedure. Through the shear force test method, the differences in adhesion strength of metal substrates with transition layers prepared by different methods were compared, and the final improvement method was determined. After that, the adhesion strength between the prepared micro–nano structure and the untreated surface polymer was verified by electron microscopy. Finally, the transition strengthening layer formed by the microstructure can be applied in most MEMS device for improving the forming ability, accuracy, and reliability of the microstructure.

## 2. Material and Methods

### 2.1. Raw Materials and Experimental Devices

A 3-inch single side polishing Si wafer with a thickness of 1000 μm (RuiCai, Suzhou, China), negative photo-resist SU-8 2150 (Kayaku, MA, USA), Propylene Glycol Methyl Ether Acetate (PGMEA, RuiCai, Suzhou, China), acetone (AR, ≥99.5%, SINOPHARM, Beijing, China), and ethanol (AR, ≥95%, SINOPHARM, Beijing, China) purchased from sinopharm are used in the following experiments, positive photoresist AZ4330 (AZ^®^ P4000 Series, Merck, Kumamoto-shi, Japan).

Magnetron sputtering equipment (MSP-400, Jinsheng, Beijing, China) and surface preparation equipment (RIE-500, Jinsheng, Beijing, China) are used in the fabricating samples.

A microscope (Imager.A2m Zeiss, Oberkochen, German), step instrument (dektak XT, Bruker Brooke dektak, Billerica, MA, USA), scanning electron microscope (ULTRA55, Zeiss, Oberkochen, German), bonding tester (PTR-1101, RHESCA, Tokyo, Japan), and AFM (Dimension Icon & FastScan Bio, Bruker, Billerica, MA, USA) are applied for characterizing the samples.

### 2.2. Problem Statement of Polymer Microstructure

Great internal stress and thermal swelling deformation are formed during the polymer process, which has a serious impact on the adhesion strength between the polymer and substrate, and on the aspect ratio and dimensional accuracy of the polymer microstructure and the electroplated structure. These problems are expressed as follows.

The swelling is mainly due to the chemical potential gradient between the polymer and the solvent molecules in the electroforming solution and the developing solution. During the swelling process, the internal volume expansion of the polymer interacts with the elastic contraction of the molecular chain, resulting in changes in the structure of the polymer and increased internal stress. In order to release the swelling phenomenon, it is necessary to increase the crosslinking reaction degree of the SU-8. Generally, appropriate or excessive photolithography (exposure and post baking) parameters are used to increase the structure replica ability.

The crosslinking reaction is the key chemistry and physics process of SU-8, which is related to the exposure dose and baking time. In this process, the internal stress is formed and increased, especially during the baking stage. The exposure dose influences the adhesion strength of the SU-8 and the substrate due to the crosslinking reaction degree and adhesive property of the SU-8. The stress is usually reduced by reducing the baking temperature or shortening the baking time, which leads to insufficient crosslinking. In this way, it is easy for swelling to occur in the development process, and in insufficient crosslinking, SU-8 is dissolved and easily leads to structural collapse.

Finally, thermally induced stress. With the increase in the crosslinking reaction degree of the SU-8, the thermal stress between the SU-8 and the substrate increases due to the influence of the thermal mismatch of the SU-8 and the substrate. In order to maintain the stability of the SU-8 structure, it is necessary to increase the adhesion strength between the SU-8 and the substrate. In the following part, the adhesion strength is improved by physical and chemical modification of the metal base without reducing the bonding degree of the SU-8 and adding additional structure.

### 2.3. Substrate Preparation

The silicon wafer was used as the basic substrate. It was firstly cleaned by standard RCA (Radio Corporation of America) procedure to remove the organic and inorganic impurities on the surface, and then a magnetron sputtering machine was used to sputter the Chromium and Cuprum (CrCu) seed layer on the surface of the cleaned substrate. The silicon with sputtered CrCu seed layer serves as the initial prepared substrate (IPS).

In order to be clearly informed about the influence of the substrate on the interfacial bonding strength, the IPS will be treated further in different ways. Cu film is electroplated on IPS to investigate the influence of electroplating film and sputtering film. Oxygen ion bombardment with different treating time is arranged to treat the substrate surface to explore the effect of textured morphology on interfacial bonding strength, since it can effectively remove organic matter and change the surface morphology with the energy base induced by adjusting the oxygen ion bombardment, which is related to the energy barrier.

### 2.4. Nano/Microstructures Activated Strengthening Layer Preparation

Furtherly, a transition strengthening layer is constructed since it can not only enlarge the contacting interfacial surface area of the SU-8 and the treated-substrate, but also increase the fasten area. Arrayed nano/microstructures are also introduced on the IPS to embed into the SU-8 microstructure to constructing transition strengthening layer.

The arrayed microstructures are fabricated by graphic array electroplating. In this method, a microstructure array was first formed by lithography, and then after resin residue cleaning by oxygen plasma treatment, electroplating was performed. Since a pushpin-like structure (shown in Figure 1) possesses a higher capacity in contacting interfacial surface area and fasten area, a pushpin-like structure array is formed by adjusting the electroplating process.

The specific process flow is as follows. First, the AZ4330 positive photoresist was spin-coated on the IPS at 1500 rpm for 30 s, then prebaked at 95 °C for 6 min. After being cooled to room temperature (~23 °C), the wafer was then exposed to 8.5 mW/cm^2^ light intensity for 7.5 s and developed for 40 s to obtain a circular hole with a depth of 5 μm and a diameter of 5 μm. 

After the lithography process was completed, the wafer was then transferred to a standard Cu electroplating process with a current density of 10 mA/cm^2^ (the plating rate about 0.2 μm per minute). Another 5 μm was plated to form the cap of the pushpin-like structure after the plating thickness was equal to the thickness of the photoresist. After removing the photoresist by immersed in acetone, the wafer was cleaned by deionized water (DI), and then dried by compressed air. After all the procedure above was completed, the Cu pushpin-like array was formed (shown in Figure 1).

### 2.5. SU-8 Microstructure Preparation

The processing of the SU-8 microstructure preparation is given in Figure 2, which mainly includes the following three parts: preparing SU-8 films, lithography, and development.

The surface free energy of the SU-8 is optimized by adjusting the gradient heating processes to prevent water vapor from condensing on its surface and forming bubbles during baking. After that, the SU-8 is spin coated on the surface of the prepared substrate and baked with the leveled hot stage. The photoresist baking adopts a multistage gradient temperature rise with 45 °C, 70 °C, and 95 °C, and a naturally cooling method to relieve internal stress caused by thermal mismatch.

Lithography was carried out with 7.5 mW/cm^2^ under 60 s. After exposure, post-exposure baking including three steps was followed, consisting of standing or 10 min at room temperature (~23 °C) to promote the uniform distribution of gradient photoacid, especially in vertical direction, then baking at 95 °C to promote the SU-8 cross-linking action, and finally using medium baking at 55 °C and also naturally cooling to room temperature to promote the SU-8 cross-linking reaction and reduce the internal stress. After that, the PGMEA solution was prepared and used for development with stirring, and the developed substrate was baked in an oven at 60 °C for 1 h to remove the residue of the solution.

### 2.6. Adhesion Strength Testing

Adhesion strength between the SU-8 and the substrates will be measured by the bonding test tool to evaluate the reliability and stability of the SU-8 process. σ (in MPa) is used as the adhesion strength and defined as Equation (1).
σ = F/A(1)
where, F represents the strength applied to the testing structure, and A is the area of bonding interface between SU-8 pillar and treated surface. The adhesion strength was measured using a PTR-110 thrust tester.

## 3. Results and Discussion

### 3.1. Influence of the Textured Morphology Induced by Sputtering and Electroplating

It is necessary to manipulate the grain size and surface topography of the initial surface in order to provide a better bonding strength [16]. This can be achieved by sputtering Cu film and electroplating Cu film on the initial substrate. The electroplated Cu film has greater grain sizes and a higher roughness than the sputtered Cu film, as shown in Figure 3A,B. The SU-8 microstructures are fabricated on the substrate with sputtered Cu film and electroplated Cu film. The interfacial adhesion strength is measured by the thrust tester, as shown in Figure 3C. This tester is destructive testing that is a fixed shear force given by the thrust tester to push down the testing sample. The test values of interfacial adhesion strength are similar, which are about 6.51 MPa and 6.57 MPa, respectively. In these conventional preparation conditions, the difference in grain size and surface morphology is not sufficient to influence the interfacial adhesion strength caused by the sputtering process and electroplating process.

### 3.2. Influence of Oxygen Ion Bombardment

With low energy oxygen ion bombardment, the surface roughness of film can be reduced, the surface is made more compact, and the external environment is prevented from damaging the internal structure, resulting in enhanced bonding at the surface. Nevertheless, it is not easy to evaluate and control the surface energy by different oxygen ion bombardment. The surface roughness of different oxygen ion bombardment is measured to analyze the influence of oxygen ion bombardment, as shown in Figure 4. It is important to note that the surface roughness of the sputtering and electroplating IPS first decreases, plotted in Figure 4, which may be due to the substrate suffering a stage of refining and flattening micro-bulge structure at the surface. This can be observed from Figure 3A-2 to Figure 5B-2 and from Figure 3B-1 to Figure 5B-1 for sputtering surface and electroplating surface, respectively. With the increment of power and time, the surface roughness is greatly increased, together with the oxidation of the Cu surface. Therefore, it is not easy to improve the adhesion strength by adjusting the oxygen ion bombardment with a small parameter variation range, Figure 4A,B. The surface topography of the Cu films of sputtering and electroplating with oxygen treatment is given in Figure 5A,B. The according adhesion strength given in Figure 5C shows that the influence of oxygen treatment on adhesion strength is complex and uncontrollable, but to a lesser extent.

Since the improvement of the surface by oxygen ion bombardment includes three aspects, as shown in Figure 6, the surface energy can be improved to increase the adhesion strength between the Cu surface and the SU-8, by cleaning the organic and inorganic pollutants attached to the surface. In addition, surface roughness may be changed to influence the effective contact area via the physical bombardment on the surface. Meanwhile, oxidation also occurs on the Cu surface, which will reduce the surface energy and the adhesion strength between the substrate and SU-8, when the oxygen plasma treatment time is prolonged. Therefore, the effect of oxygen treatment on adhesive strength is very complex, and the adjustment range is very narrow, which limits the repeatability and universality of the microfabrication processes.

### 3.3. Influence of Pushpin-like Reinforced Microstructure

The SU-8 microstructure is fabricated with a pushpin-like reinforced microstructure, as shown in Figure 7A-1. The adhesion strength is carried out, and the surface topography of the SU-8 is given in Figure 7A-2; the failure model is similar to that of the normal SU-8 exhibited in Figure 7B. The adhesion strength of the reinforced SU-8 reaches 15.42 MPa, which enhances it about 2.37 times, given in Figure 7C.

First, the pushpin-like nano/microstructure array can effectively improve the contact area between the polymer and the substrate. The bottom of the pushpin is the area of the original substrate, and the top is the surface area of the reinforced structure, given in Figure 8. Through calculation, it can be seen that the surface area of the reinforced structure is larger than that of the planar structure, and the surface area obtained by the arc rotation with the cylinder center as the rotation axis and the radius *r* is added. Its expression is shown as Equation (2).
(2)$$S_{arc}=\frac{\pi^{2}ra}{2}+2\pi r^{2}$$
where, *S_arc_* is the area of the increased arc, *r* is the height of isotropic plating, and *a* is the root diameter of the pattern. In this way, the contact area between the SU-8 and the substrate is greatly increased with numbers of nano/microstructures to improve the adhesion strength between SU-8 and the substrate.

In addition, the pushpin-like nano/microstructure array also introduces the mechanical interlocking effect, as shown in Figure 1, and the SU-8 among the pushpin-like nano/microstructure array is imitated. Thus, the adhesion strength between the SU-8 and the pushpin-like nano/microstructure array can be enhanced furtherly. In a word, the influence of the pushpin-like nano/microstructure array improves the adhesion strength via two aspects.

As shown in Figure 9, the pushpin-like reinforced microstructures keep good structural characteristics, and their top surfaces possess good uniformity and certain roughness; the pushpin-like structures are built up well to achieve a mechanical interlocking effect, which guarantees the amplitude of reinforcement. The reinforced microstructures can arrange one or more layers, which depends on the dimensions of the SU-8 and reinforced microstructure, resulting in the foundation for adjustable adhesion strength.

### 3.4. SU-8 Microstructure

Independent structures, gradually increasing adjacent periodic structures, and adjacent structures with large aspect ratios are typical structures in SU-8 applications. Therefore, three typical structures—including hollow square microstructures, gradually increasing adjacent periodic microstructures, and slender strip microstructures—are designed and prepared for comparison and verification. The microstructures of different control groups are fabricated with the same micro-fabrication processes, parameters, and equipment, and characterized with SEM.

Most of the hollow square SU-8 microstructure with a small size slipped away without leaving any trace—only two unites left, shown in Figure 10A-1. Corresponding to this, all of the SU-8 array composed of 6 × 6 units still firmly stood by, introducing the pushpin-like array to form the strengthening layer, shown in Figure 10A-2. In terms of the gradually increasing adjacent periodic SU-8 microstructures, all of them drifted away from the initial location, given in Figure 10B-1. Both of the units still firmly stood by applying a pushpin-like array to enhance the adhesion strength, exhibited in Figure 10B-2. As revealed in Figure 10C-1, the slender strip SU-8 microstructures were glued together, while the controlled group microstructures—which were in an embedded pushpin-like array—still firmly stood with a certain distance from each other, displayed in Figure 10C-2.

The fabricated multi-layer SU-8 microstructure formed in a large area is given in Figure 11A-1, which is about 65 × 65 mm^2^ in area. It is made of 9 × 6 unites, shown in Figure 11A-2–A-3. The fabricated multi-layer SU-8 microstructure is strong enough for serving as the sacrificial layers with no warpage and stripping. 

## 4. Conclusions

In order to improve the adhesion between the polymer and metal substrate and enhance the uniform of morphology, this study uses micro machining technology (sputtering, electroplating, oxygen ion treatment, etc.) to texture the nano/microstructure to form a transition strengthening layer. The adhesive strength between the SU-8 microstructure and various activated substrates was verified and measured by the thrust tester. To obtain more accurate measured results, the SU-8 microstructures were prepared with the same photolithography parameters and the morphology of the surface was characterized in detail. The experimental results show that an SU-8 with an activated pushpin-like reinforce microstructure array has a highly stable adhesive ability and adhesive strength. The reliability and feasibility of the new method were verified by preparing three typical SU-8 microstructures (hollow square microstructure, gradually increasing adjacent periodic microstructure, slender strip microstructures) and large-area SU-8 microstructures. Compared with the traditional design, this method effectively avoids drift and gluing. The main results are concluded as the following:(1)Based on the pushpin-like reinforce microstructure array, the effective contact area is enlarged, and the mechanical interlocking effect is introduced.(2)The adhesive strength of SU-8 with pushpin-like reinforce microstructure array was improved about 2.37 times compared with the sputtering method.(3)Based on the pushpin-like reinforce microstructure array, the drifting and gluing phenomenon can be alleviated.(4)The feasibility of the transition strengthening layer and repeatability and universality of the microfabrication processes has been evaluated by the application of pushpin-like reinforce microstructure array in three typical SU-8 microstructures and large-area SU-8 microstructures.

Furtherly, as a universal method, the pushpin-like reinforce microstructure array can be applied to the surface of other materials with an adaptive manufacturing method. In addition, other specific adhesion-enhanced methods could be used together to obtain a firmer binding to a specific interface. It will be promising to expand the polymer microstructures in MEMS by solving the poor interface bonding between the polymer and Si or metal.

## Figures and Tables

**Figure 1 polymers-15-01015-f001:**
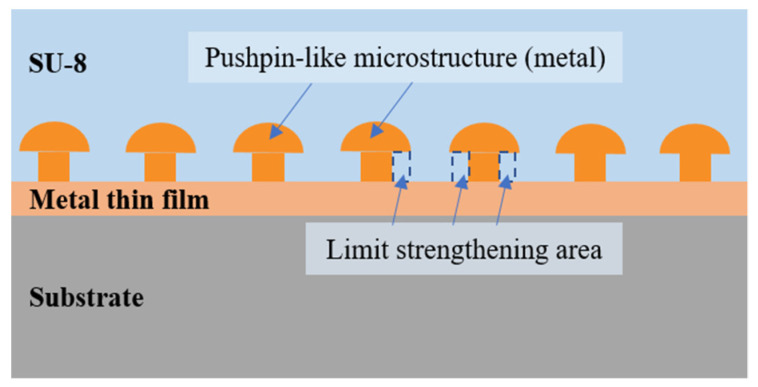
Scheme of enhanced pushpin-like structure array.

**Figure 2 polymers-15-01015-f002:**
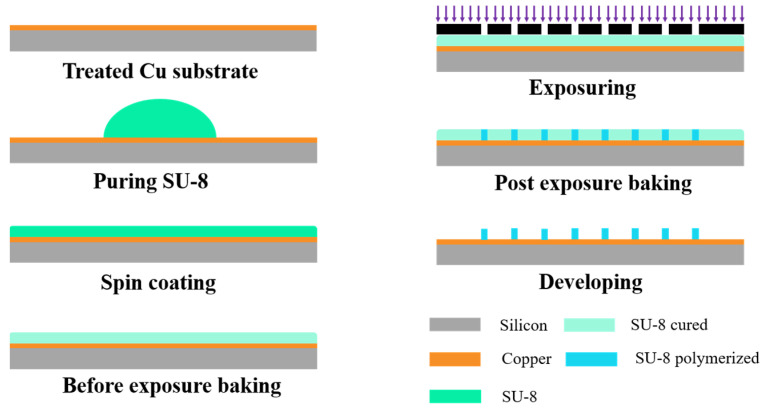
Processing flow diagram of SU-8 microstructure.

**Figure 3 polymers-15-01015-f003:**
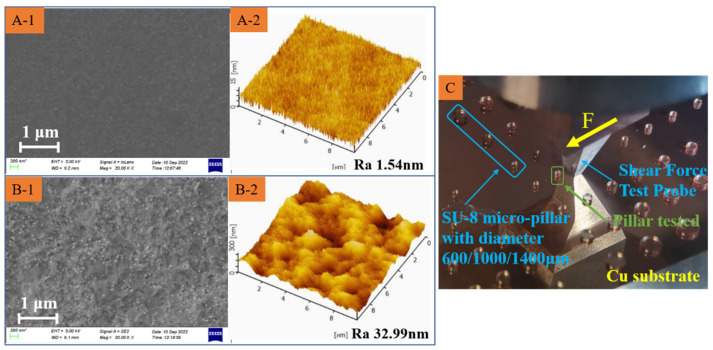
Characterization of Material Roughness and Morphology by AFM and SEM, sputtered Cu (**A-1**,**A-2**); electroplated Cu (**B-1**,**B-2**). (**C**) thrust tester.

**Figure 4 polymers-15-01015-f004:**
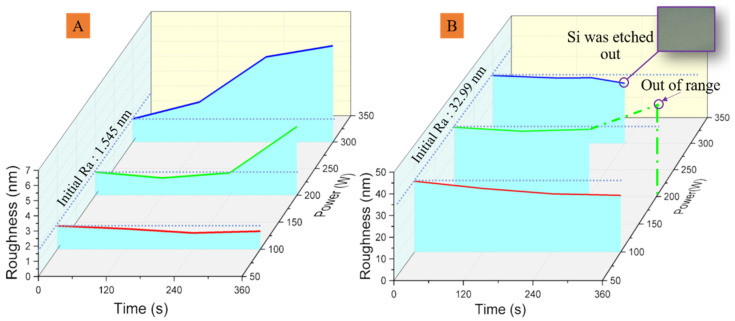
Surface roughness for different oxygen plasma treatment measured with AFM, (**A**) sputtering Cu and (**B**) electroplating Cu.

**Figure 5 polymers-15-01015-f005:**
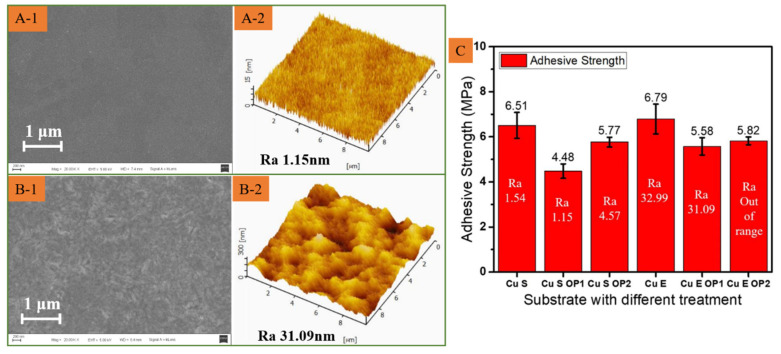
SEM and AFM of sputtered Cu (**A-1**,**A-2**); electroplated Cu (**B-1**,**B-2**) with oxygen ion bombardment. (**C**) adhesion strength test results.

**Figure 6 polymers-15-01015-f006:**
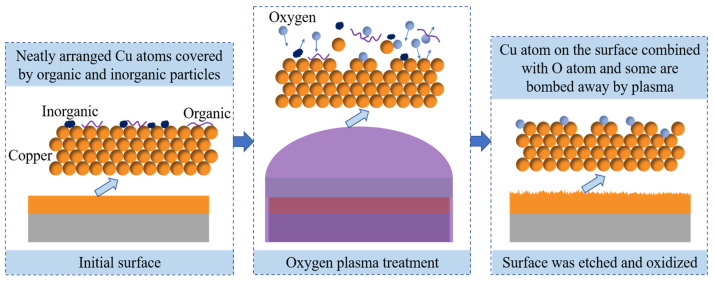
Physicochemical change form of interface during oxygen plasma treatment.

**Figure 7 polymers-15-01015-f007:**
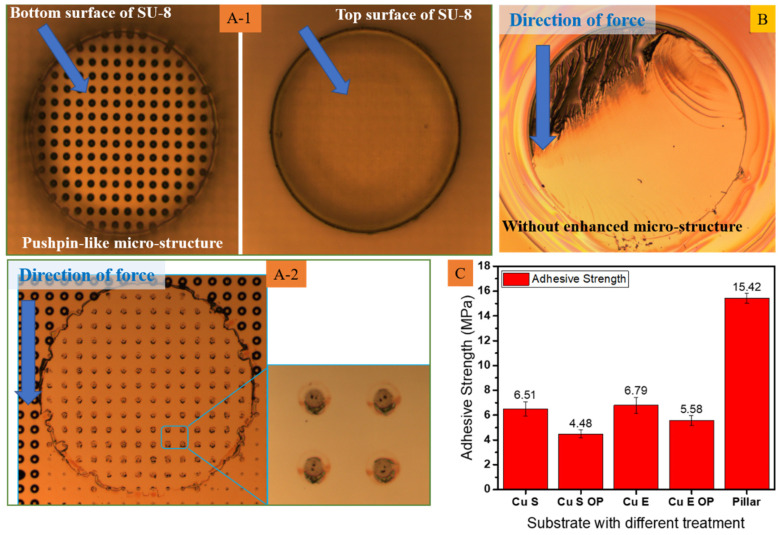
Surface topography of SU-8 with pushpin-like reinforce microstructure array (**A-1**) before and (**A-2**) after thrust test. (**B**) Surface topography of normal SU-8. (**C**) Adhesion strength.

**Figure 8 polymers-15-01015-f008:**
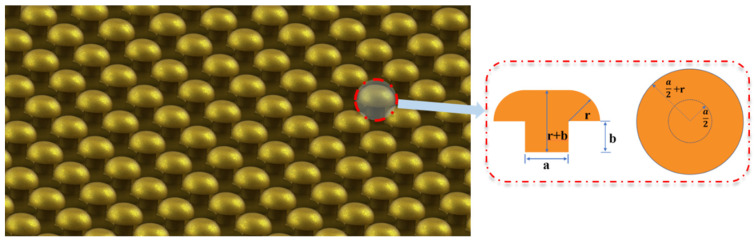
Pushpin-like reinforce microstructure array.

**Figure 9 polymers-15-01015-f009:**
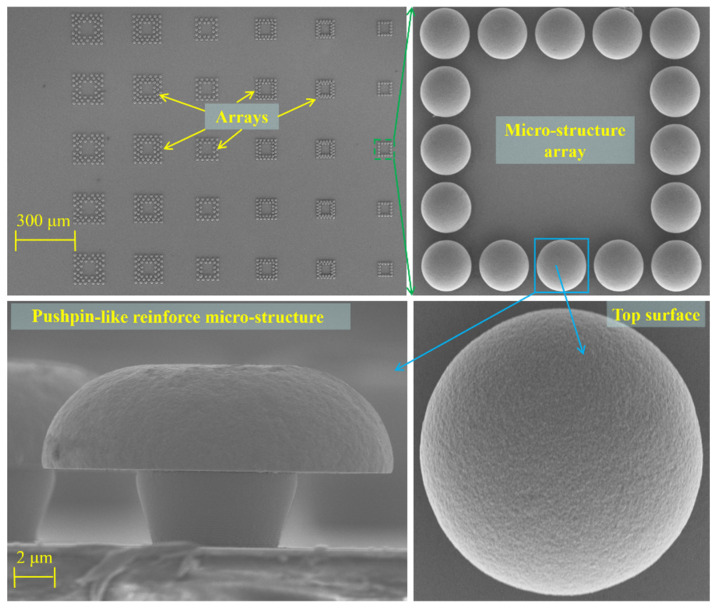
One kind of pushpin-like microstructure profile characterized by SEM.

**Figure 10 polymers-15-01015-f010:**
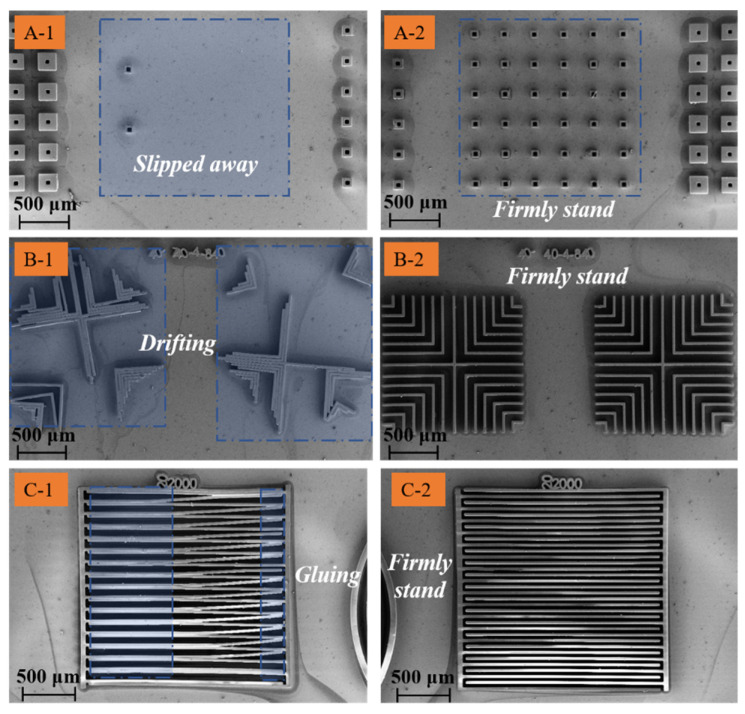
SEM of different SU-8 microstructures IPS (**A-1**–**C-1**) and with strengthening layer on IPS (**A-2**–**C-2**).

**Figure 11 polymers-15-01015-f011:**
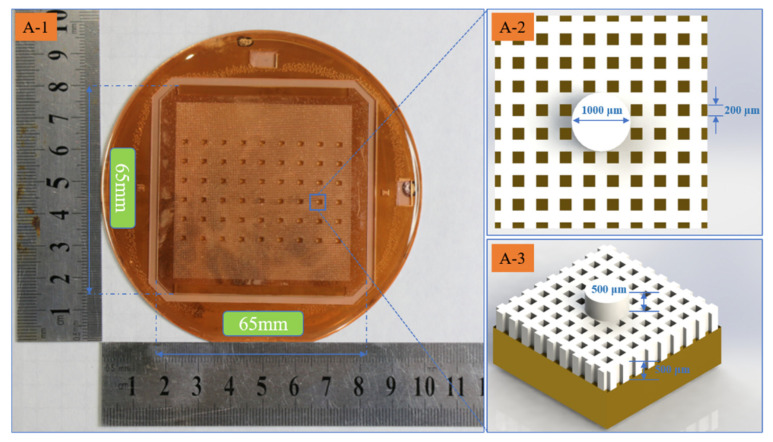
Structure diagram of multi-layer SU-8 microstructure formed in large area. (**A-1**) Surface topography of overall structure, (**A-2**) surface topography, and (**A-3**) 3D structure of unite.

## Data Availability

The data presented in this study are available on request from the corresponding author.

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
