# Peer review of "Preparation of Highly Stable Polymer Microstructure with Enhanced Adhesion Strength by Pushpin-like Nano/Microstructure Array"

_polymers, 2023, doi:10.3390/polym15041015_

Round 1
Reviewer 1 Report
The paper is interesting, but it is also puzzling for me. The authors do not explain in detail how they produce the pushpin-like structures; the parameters they used for the lithographic process that could allow to reproduce this study are not provided. A deeper exposition of the process should be provided to allow the publication of the paper.
Apart for that, some other points are:
- Citations should be placed in parentheses [].
- what is PI? First page, 5 lines from the bottom.
- state what SU-8 is the first time you use the acronym
- in general, define all acronyms before using them.
- please, give the exact model of the instruments
- section 2 could be shortened and placed in the following section
- section 3 should be “Materials and methods”
Reviewer 2 Report
The paper entitled “Preparation of highly stable polymer microstructure with en-hanced adhesion strength by pushpin-like nano/micro-struc-ture array” by Yongjin Wu, Guifu Ding, Yuan Zhu, b, Yan Wang, Rui Liu, Yunna Sun submitted for publication in Polymers achieves a new polymeric material with better adhesion strength, microstructurally studied.
The length of the paper is short, but the article is concise and easy to understand. Of course, there is some section that squeaks like the references, since recent publications on adhesion strength are missing (DOI:
10.1016/j.surfcoat.2022.128752), publications of the methodology used here, and especially in competition with other works, looking for reviews (DOI: 10.3390/polym14142856) especially in the field of applications in curing/coating, biofilms (DOI:
10.1016/j.chemosphere.2022.135740) and stretching the thread, of composites (DOI: 10.1016/j.compositesa.2022.107275), also looking for related to gel formation (DOI: 10.1021/acs.chemmater.2c01346). From the point of view of the format there is no concern, but also in the references there are most errors, from commas before the year of publication, between authors ... references like the 9 that seems to be in the middle firm and explicitly expressed VOL. of volume. The 8 also does not understand the format of the page. 11 has full names... 18 has ellipsis... And then how the references are cited in the text is more than surprising, with two hyphens between the numbers, what is reference 0 by the way?... or that is, redo everything, rethinking it with meaning and completeness.
Then linked to what was mentioned in the previous paragraph, the results are understandable but there is a lack of context to compare them to.
Right now I cannot decide if the work should go ahead or not, and I am waiting for a conviction in the comments/texts added by the authors, and that's why I'm asking for Major Revisions.
Round 2
Reviewer 1 Report
The authors sufficiently improved their manuscript
Author Response
Thank you.
Reviewer 2 Report
After reading the new draft, I keep my first report because the authors considered that the introduction included all the necessary work in the field… if this is the case this means that the work is that not interesting because removing the methodological references the number of references is ridiculous, and overall just 25. Only the explanation why the structures that were discussed are chosen is satisfactory, but maybe my comment was confusing: the authors should compare to any previous work with similar or different polymers to understand where we are and how useful those results could be.
One clear example that references have a problem is that it appears in the first paragraph of the introduction “[Error! Reference source not found.-” and actually, references start from reference 4? Thus, 1-3 are useless? Recheck all of them since the problems still are there (not as before), take for instance the journal titles are not in itallics, missing free spaces… (refs. 11, 12, 14…)
Thus, since the comparison with respect to other polymer materials and the past issues remain, I keep my decision in Major revisions.
Round 3
Reviewer 2 Report
In this second revision the answers are detailed, and the photos certainly help too. And at least for me, I found it very interesting to see how the assembly is, and the answer of "pulling vs shearing" is quite remarkable.
And then we would enter the discussion of science and expand horizons. Since the beginning of the revisions I believe I remember 20-25 references now the article has improved in this aspect. I may think that I may have a wrong view when I think that the introduction must have clicks, some perspective comment to attract the interest of readers, new readers... and I could be wrong. And to check that I could be wrong I went to do a self-personal exercise, to look at the impact of my articles, then of the main author of the article under review, homogenizing only for the articles from 2014, and also the last 5 years and also by article. In summary, from the beginning 300 citations, a figure that results in roughly 5 citations per article, and for the sake of ethics I will not put my numbers. And so I am really surprised that yes, putting “links, "cheats" helps to expand the interest of the articles. For me it has served to unite subjects, to put a solution of one work into others that are totally different, but interrelated by the materials, molecules used, get collaborative projects...
I also understand why sometimes there are reviewers who simply reject articles saying: "lack of interest", "too specialized"... and this strategy of referring to close fields could help. Anyway, it is advice. For me the work is good here... but I see difficult that traps the interest of readers. I repeat, maybe I am wrong… but if after 2-3 years any of my papers has the minimum of 10 citations… I think the paper was not catchy enough and that it is my fault as a scientist.
Author Response
Thank you for your advice.